# Crosstalk between NLRP12 and JNK during Hepatocellular Carcinoma

**DOI:** 10.3390/ijms21020496

**Published:** 2020-01-13

**Authors:** Shahanshah Khan, Hasan Zaki

**Affiliations:** Department of Pathology, UT Southwestern Medical Center, Dallas, TX 75390, USA; shahanshah.khan@utsouthwestern.edu

**Keywords:** JNK, NLRP12, hepatocellular carcinoma, inflammation

## Abstract

Hepatocellular carcinoma (HCC), a leading cause of cancer-related death, is initiated and promoted by chronic inflammation. Inflammatory mediators are transcriptionally regulated by several inflammatory signaling pathways, including nuclear factor kappa B (NF-κB) and mitogen-activated protein kinase (MAPK). cJun N-terminal kinase (JNK), a member of the MAPK family, plays a central role in HCC pathogenesis. Pathogen-associated molecular patterns (PAMPs) activate JNK and other MAPK upon recognition by toll-like receptors (TLRs). Apart from TLRs, PAMPs are sensed by several other pattern recognition receptors, including cytosolic NOD-like receptors (NLRs). In a recent study, we demonstrated that the NLR member NLRP12 plays a critical role in suppressing HCC via negative regulation of the JNK pathway. This article briefly reviews the crosstalk between NLRP12 and JNK that occurs during HCC.

## 1. Introduction

Hepatocellular carcinoma (HCC) is the most common type of primary liver cancer [1,2]. With only a 18% survival rate in 5 years, HCC ranked as the second highest death-causing cancer, after pancreatic cancer [3]. Mutations in various oncogenes, including *CTCNB1*, *WNT*, *AXIN*, *TP53*, *CCND1*, and *CDKN2A* are commonly found in HCC [4,5]. The primary cause for sporadic mutations and neoplastic transitions of parenchymal cells is chronic injury and inflammation induced by hepatitis B and C virus (HBV and HCV) infections, chronic alcohol consumption, and drug toxicity [2,6]. Irrespective of etiologies, inflammation plays a central role in the induction and promotion of HCC. For example, inflammatory mediators cause DNA damage, induce mutations, trigger cell death, and promote proliferation of neoplastic hepatocytes [7,8].

The major pathways regulating inflammation in the liver include nuclear factor kappa B (NF-κB) and mitogen-activated protein kinase (MAPK) [8,9,10,11,12,13]. These inflammatory pathways are activated by pathogen-associated molecular patterns (PAMPs), danger-associated molecular patterns (DAMPs), cytokines, growth factors, and stress. Among diverse stimuli, PAMPs are the most potent activator of NF-κB and MAPK pathways. Because of its close anatomical connection with the intestine, the liver is constantly exposed to gut microbiota-derived PAMPs, suggesting that PAMPs constitute a critical player in inflammatory responses and HCC pathogenesis as well [14]. Clinical evidence showing increased endotoxins in patients with chronic liver disorders further underscores the link between chronic liver inflammation and gut-derived PAMPs [15,16,17,18,19,20]. PAMPs are sensed by pattern recognition receptors (PRRs), such as toll-like receptors (TLRs), NOD-like receptors (NLRs), RIG-I-like receptors (RLR), AIM2-like receptors (ALR), and several other cytosolic receptors for nucleic acids [21,22]. Involvement of these PRRs in the pathogenesis of HCC is increasingly evident [23,24,25,26,27]. We recently investigated the role of NLRP12, an NLR member, in HCC pathogenesis [28]. This study demonstrated that *Nlrp12*^-/-^ mice are highly susceptible to hepatocellular carcinoma [28]. Increased HCC pathogenesis of *Nlrp12*^-/-^ mice was associated with increased activation of the c-Jun N-terminal kinase (JNK) signaling pathway. Here, we will review our findings focusing on the link between NLRP12 and JNK in the regulation of HCC.

## 2. JNK in HCC Pathogenesis

The Ras/Raf/MAPK pathway is one of the major pathways associated with HCC [11,29]. Three major MAPK pathways are ERK, P38, and JNK. JNK is activated by dual phosphorylation of the tripeptide motif (Thr-Pro-Tyr) by upstream MAPK kinases MKK4 and MKK7, which are activated by a cascade of kinase signaling [30]. PAMPs activate the JNK pathway following recognition by TLRs. In addition, cytokines (e.g., TNFα, IL-1α), growth factors, mitochondrial stress, and environmental stress can activate JNK [30,31]. Activated JNK phosphorylates AP-1 family transcription factors, such as cJun, cFos, JunB, JunD, ATF2, and non-AP-1 transcription factors, including p53, cMyc, FOXO4, STAT1, and STAT3 [30,32]. These transcription factors regulate a myriad of genes involved in inflammation, proliferation, cell death, and oncogenesis.

Since the JNK pathway plays a central role in maintaining homeostasis in the liver, its dysregulation is associated with inflammatory liver disorders, including steatosis, fibrosis, cirrhosis, and HCC [29,33,34]. JNK induces and activates cJun and cMyc, oncogenic transcription factors that are highly expressed in the HCC tissue [35,36,37,38,39,40]. As such, deletion of cJun in hepatocytes dramatically reduced the number and size of liver tumors induced by diethylnitrosamine (DEN), a hepatotoxin and carcinogen, in mice [41]. Reduced tumor burden in cJun-deficient mice was associated with higher expression of tumor suppressors p53 and p21 [39,41]. Consistently, JNK deficiency or its inhibition abrogates proliferation of tumor cells in the liver [31,34,42]. Compelling evidence suggest that JNK plays an essential role in hepatocyte proliferation and liver regeneration [43,44]. JNK promotes proliferation of hepatocytes via induction of molecules involved in cell cycle progression, such as Cyclin D1 and cMyc, as well as suppression of p53 and p21, which induce apoptosis and inhibit proliferation [39,44,45,46].

Because of its tumor-inducing potential, activation of JNK in the liver needs to be maintained tightly, and such a regulation involves other inflammatory pathways and molecules. Mice deficient in IκB kinase β (IKKβ) in hepatocytes developed increased HCC following the administration of DEN [8]. This finding was unexpected, since transcription factor NF-κB, which is activated by IKKβ, is the major regulator of inflammation. However, further investigation revealed that a higher tumor burden in hepatocyte-specific IKKβ-deficient mice was associated with increased activation of JNK [8]. Similarly, embryonic fibroblasts and fetal hematopoietic cells from p38α-deficient mice showed hyper-proliferation resulting from increased activation of JNK [47,48]. Also, mice deficient in p38α in hepatocytes exhibited increased HCC pathogenesis, which was accompanied by higher activation of JNK [7,48]. All these provide clear evidence that JNK cross-talks with other inflammatory pathways, including NF-κB and p38.

Among three isoforms of JNK (JNK1, JNK2, and JNK3), liver expresses JNK1 and JNK2. However, DEN-induced HCC development was significantly reduced in *Jnk1*^-/-^ mice, but not *Jnk2*^-/-^ mice [13,47]. Furthermore, human HCC tissue exhibited increased JNK1 activation, while JNK2 remained similar to normal liver, suggesting that JNK1 is the main player in HCC pathogenesis [13]. In agreement with this, inhibition of JNK with SP600125, a chemical inhibitor, was seen to suppress DEN-induced HCC development [40]. While it is overwhelmingly supported that JNK activation promotes cellular proliferation, animal studies documented an intriguing phenomenon that JNK-mediated HCC pathogenesis is associated with increased apoptosis [31]. Given that proliferation and apoptosis are two opposite biological processes, and tumorigenesis is considered a defect in cell death, how increased apoptosis promotes tumorigenesis is intriguing. These opposing processes in HCC pathogenesis was explained by the fact that hepatocyte death triggers compensatory proliferation of surviving hepatocytes, leading to the induction and promotion of HCC [7]. JNK-mediated hepatocyte death is accompanied by the release of proinflammatory mediators, such as IL-1α, TNFα, and reactive oxygen species (ROS), which stimulate resident macrophages (Kupffer cells), infiltrated macrophages, and dendritic cells to express proinflammatory cytokines, such as IL-6, IL-α, IL-β, and TNFα [7]. These cytokines, along with other growth factors released by myeloid cells, promote the proliferation of neoplastic hepatocytes [10,13,24,49,50,51].

Although most in vitro and in vivo studies depicted JNK as a promotor for HCC, it is surprising that mice double-deficient in Jnk1 and Jnk2 in hepatocytes are susceptible to HCC [52]. In contrast, hematopoietic-specific deletion of Jnk1 and Jnk2 protects animals against HCC [52]. These apparently conflicting observations were explained by the fact that the complete absence of JNK leads to hepatocyte death, promoting hyperactivation of JNK in Kupffer and other myeloid cells and, in turn, induction of inflammatory molecules, which trigger compensatory proliferation of transformed hepatocytes [52]. Considering the crosstalk between JNK and other inflammatory signaling pathways, it is intriguing whether NF-κB, ERK, and P38 become hyperactivated in hepatocytes deficient in both JNK1 and JNK2. Such hyperactivation of other inflammatory pathways in the absence of JNK may promote HCC.

## 3. NOD-Like Receptors

The innate immune response against pathogens is initiated with germline-encoded PPRs [53,54,55]. Among several families of PRRs, TLRs and NLRs are most discussed [56]. While TLRs are located on the cell surface and the endosomal compartment, NLRs are present in the cytosol [55,57]. NLRs recognize both PAMPs and DAPMs, such as ATP, uric acid, hyaluronan, and other host-derived cellular and metabolic products [58]. NLR-mediated recognition of PAMPs and DAMPs results in the activation of downstream signal transduction pathways, leading to either induction or suppression of inflammatory responses [58,59,60]. Structurally, NLRs share the typical tripartite structural domain organization, including an N-terminal effector domain, a central nucleotide-binding NATCH domain, and a C-terminal LLR domain responsible for the recognition of pattern molecules [22,59]. At least 22 NLR members have been identified in humans, and they are divided into four subfamilies based on their N-terminal effector domains, including: (1) NLRA—contains an acidic transactivation domain; (2) NLRB—having a baculoviral inhibitory repeat-like (BIR) domain; (3) NLRC—possessing caspase activation and recruitment domain (CARD); and (4) NLRP—pyrin domain containing NLRs. Interestingly, in contrast to TLRs, NLRs play diverse roles, including activation of NF-κB and MAPK pathways, formation of the inflammasome, suppression of inflammatory signaling pathways, and transcriptional regulation of genes [22,61]. Notably, NLRs often exhibit a cell type-specific function, and several NLRs share overlapping functions. Because of their key roles in innate immunity and inflammation, NLRs have been implicated in infectious, auto-immune, and inflammatory disorders including cancer [61,62,63,64].

## 4. NLRP12 and Inflammatory Disorders

NLRP12 belongs to the NLRP subfamily of NLRs. Like other NLRP family proteins, NLRP12 is composed of an N-terminal pyrin domain (PYD), central nucleotide binding domain (NBD), and a C-terminal leucine-rich repeat (LRR). NLRP12 is abundantly expressed in myeloid cell lineage, including macrophages, dendritic cells, monocytes, and neutrophils [65,66]. An initial study found that NLRP12 interacts with ASC in vitro, predicting its role in inflammasome activation [67]. However, the role of NLRP12 in the activation of the inflammasome has not been observed in diverse pathophysiological contexts, except infection caused by *Yersinia pestis* [68]. Most other studies described NLRP12 as a negative regulator of inflammatory responses [66,69,70,71]. Missense mutations in NLRP12 have been identified in patients with atopic dermatitis and periodic fever syndrome [72,73,74]. Mice deficient in Nlrp12 are highly susceptible to chemically induced colitis and colorectal tumorigenesis [66,69]. Increased inflammation and tumorigenesis of *Nlrp12*^-/-^ mice are associated with higher inflammatory responses and activation of NF-κB and ERK pathways [66,69]. Supporting this in vivo observation, bone marrow-derived macrophages and dendritic cells of *Nlrp12*^-/-^ mice are hyper-responsive to TLR ligands, such as LPS, Pam3, and Poly I:C [66,69]. Cytokines, chemokines, and other inflammatory mediators contribute to host defense against bacterial infection. Consistently, higher inflammatory responses in *Nlrp12*^-/-^ mice during *Salmonella* Typhimurium infection helped resolve the infection [70]. A recent study demonstrated that NLRP12 dampens antiviral immune responses; however, such a regulation involved the RIG-I pathway but not NF-κB and MAPK [75], suggesting that NLRP12 may regulate inflammatory response and host immunity in multiple ways.

While most studies found NLRP12 to inhibit NF-κB and MAPK pathways in myeloid cells, increasing evidence suggests that NLRP12 regulates these pathways in other cell types as well. T cells of *Nlrp12*^-/-^ mice are highly responsive to antigen immunization. Thus, *Nlrp12*^-/-^ mice develop atypical experimental autoimmune encephalomyelitis (EAE) due to increased production of Th2 cytokine IL-4 into the central nervous system [71]. Transfer of naïve *Nlrp12*^-/-^ T cells into immunodeficient *Rag*^-/-^ mice also elicited exacerbated colitis [71]. The role of NLRP12 in attenuating these inflammatory disorders was explained by the fact that NLRP12 downregulates NF-κB and ERK activation in T cells [71]. Deficiency of NLRP12 also promotes proliferation of osteocytes, hepatocytes, and microglia [28,76,77]. In the intestine, NLRP12-mediated regulation of inflammatory responses may shape gut microbiota composition [78,79]. Recent studies suggest that altered gut microbiota predisposes susceptibility to colitis and obesity in *Nlrp12*^-/-^ mice [78,79]. While these observations are interesting, gut microbiota composition can be modulated by other factors, including the animal facility environment, geographical location of the laboratory, mouse handling practices, diet, and so forth. Thus, gut microbiota-dependent disease phenotypes of *Nlrp12*^-/-^ mice should be validated by independent studies from other laboratories.

## 5. NLRP12 Suppresses Hepatocellular Carcinoma

Given that chronic inflammation is a major driver for HCC and NLRP12 negatively regulates inflammatory responses, it is intriguing whether NLRP12 plays any role in HCC. In a recent study, we investigated the role of NLRP12 in HCC using mouse models in which tumors were induced by DEN or DEN plus carbon tetrachloride. In both experimental settings, *Nlrp12*^-/-^ mice developed significantly higher numbers of and larger tumors compared to WT mice [28]. This observation suggests that NLRP12 plays a protective role against HCC. The cancer genomics database suggests that *NLRP12* is altered in about 2% of HCC patients [28]. Although *NLRP12* is not a major cancer suppressor gene, its expression and activation status may regulate HCC pathogenesis.

Increased HCC pathology in *Nlrp12*^-/-^ mice was associated with higher expressions of the HCC marker Afp, inflammatory cytokines, and chemokines, including IL-6, TNFα, Cxcl1, Cxcl2, and Ccl2, protooncogene cJun, cMyc, and Cyclin D1, and reduced expression of p21 [28]. IL-6 and TNFα are critical players in HCC pathogenesis with their functions in cellular proliferation and cell death [7,8,80,81,82]. These two cytokines were found to be elevated in the liver of HBV infected patients, further supporting their association in HCC pathogenesis [83,84]. In addition to these pro-inflammatory cytokines, chemokines that recruit macrophages and other myeloid cells in the tumor microenvironment play important roles in HCC [7,8,82]. Inflammatory mediators produced by Kupffer cells and other immune cells contribute to the development of steatosis, fibrosis, and cirrhosis in the liver [6,85,86]. Higher steatosis and fibrosis in DEN-treated *Nlrp12*^-/-^ mouse livers, therefore, reflect an overall hyperinflammatory response [28]. Notably, inflammatory and proliferative molecules were not dysregulated in healthy *Nlrp12*^-/-^ livers [28], indicating that NLRP12 suppresses those tumor-promoting mediators in the context of liver injury.

## 6. NLRP12 Negatively Regulates JNK Activation in the Hepatocyte

As discussed above, inflammatory signaling pathways, including NF-κB, ERK, P38, JNK, and STAT3, regulate inflammatory responses and tumorigenesis. Since NLRP12 has been shown to downregulate the activation of NF-κB and ERK, these pathways were expected to be hyperactivated in *Nlrp12*^-/-^ livers. Interestingly, *Nlrp12*^-/-^ HCC showed higher JNK activation, but not NF-κB and ERK [28]. This observation suggests that NLRP12 regulates different inflammatory pathways in a cell type-specific manner. Indeed, higher activation of JNK was seen only in *Nlrp12*^-/-^ hepatocytes; there was no major difference in JNK activation in Kupffer cells and hepatic stellate cells isolated from wild-type and *Nlrp12*^-/-^ mouse HCC [28]. The hepatocyte intrinsic function of NLRP12 in regulating JNK was confirmed by in vitro biochemical assays. Primary hepatocytes from healthy *Nlrp12*^-/-^ mice exhibited increased activation of JNK and expression of cJun, cMyc, and Ccnd1 upon stimulation with LPS and other TLR ligands, e.g., Pam3 and PGN [28]. Knockdown of NLRP12 in the human HCC cell-line HepG2 provided similar results [28]. Corroborating with these data, JNK activation and expression of JNK downstream molecules were markedly reduced upon overexpression of NLRP12 in HepG2 cells [28]. Overall, these studies strongly imply that NLRP12 is a critical negative regulator of the JNK pathway in the liver.

## 7. NLRP12 Regulates Hepatocyte Proliferation via JNK

A unique feature of the liver is its regeneration capacity. Liver that is subjected to partial hepatectomy regains its complete mass within 7 to 10 days [39,44]. Liver injury or cytotoxicity leads to rapid proliferation of surviving hepatocytes to maintain liver homeostasis [7,87,88]. While compensatory proliferation of hepatocytes is essential for liver homeostasis, this process also triggers HCC during chronic inflammation and injury, or in the presence of mutagens. In agreement with this paradigm, we observed increased apoptosis as well as proliferation in *Nlrp12*^-/-^ liver upon DEN-induced HCC induction [28]. Notably, healthy WT and *Nlrp12*^-/-^ livers did not show any difference in cell death and proliferation [28]. The role of NLRP12 in hepatic cell death was examined in vitro using primary hepatocytes from WT and *Nlrp12*^-/-^ mice. The proliferation rate of hepatocytes was significantly higher in *Nlrp12*^-/-^ mice, although no major difference in cell death was observed between WT and *Nlrp12*^-/-^ hepatocytes [28]. Notably, hepatocyte proliferation was increased in the presence of LPS, further supporting the concept that inflammatory pathways enhance proliferation of hepatocytes [28]. Thus, higher apoptotic death of hepatocytes following administration of DEN in the *Nlrp12*^-/-^ liver was due to extrinsic factors, such as higher abundance of reactive oxygen and nitrogen species, cytotoxic cytokines (e.g., TNFα), and proapoptotic molecules in the tumor microenvironment. Higher proliferation of *Nlrp12*^-/-^ hepatocytes was associated with increased expression of pro-proliferative molecules, including cMyc, cJun, and Ccnd1. The JNK inhibitor dramatically reduced proliferation rate of hepatocytes and expression of cJun, cMyc, and Ccnd1, suggesting the key role of JNK in the hyperproliferative nature of *Nlrp12*^-/-^ hepatocytes [28].

## 8. NLRP12 Suppresses Gut Microbiota-Mediated Activation of JNK in the Liver

JNK can be activated by PAMPs, as well as host-derived factors, such as TNFα, ROS, and stress [30,82,89,90]. However, microbial pattern molecules, such as LPS, MDP, and PGN, are more potent as activators of NF-κB and MAPK pathways compared to endogenous stimuli. Increasing evidence suggests that gut-derived microbial products play pivotal roles in HCC pathogenesis [14]. Mice raised in germ-free (GF) conditions do not develop DEN-induced HCC [24]. In agreement with this, mice deficient in TLR4 which sense LPS are less susceptible to HCC [23,24]. Depletion of gut microbiota with antibiotics also dramatically reduces tumor growth [24]. It appears that higher JNK activation in *Nlrp12*^-/-^ tumors is due to increased activation of TLR pathways by LPS or other PAMPs. Consistently, *Nlrp12*^-/-^ mice treated with antibiotics did not develop visible tumors [28]. In fact, microbial products are circulated to the liver through the portal vein after crossing the intestinal epithelial barrier. Therefore, our data further strengthens the growing body of evidence that gut-derived microbial products activate JNK and other inflammatory pathways in the liver, promoting HCC (Figure 1).

## 9. Concluding Remarks

Gut microbiota-derived PAMPs play a central role in HCC pathogenesis. Both parenchymal and non-parenchymal cells express TLRs, and therefore, may respond to PAMPs, leading to the activation of JNK and other inflammatory pathways. Therefore, suppression of this gut-liver axis of JNK pathway may constitute an effective means for inhibiting HCC pathogenesis. Our recent study points to NLRP12 as an important molecular brake of this signaling axis and HCC pathogenesis. It is interesting that NLRP12 regulates JNK but not NF-κB and ERK in hepatocytes. Whether NLRP12 negatively regulates the JNK pathway in a manner similar to its inhibition of NF-κB and ERK in myeloid cells remains unknown. Previous studies have documented that NLRP12 interacts with IRAK1 to suppress canonical NF-κB, and NIK and TRAF3 to inhibit non-canonical NF-κB in myeloid cells [69,91,92]. It would be interesting to see if NLRP12 interacts with these or other signaling adapters to attenuate JNK activation. Identification of kinases or interacting partners involved in NLRP12-mediated inhibition of the JNK pathway may facilitate finding new therapeutics for HCC.

## Figures and Tables

**Figure 1 ijms-21-00496-f001:**
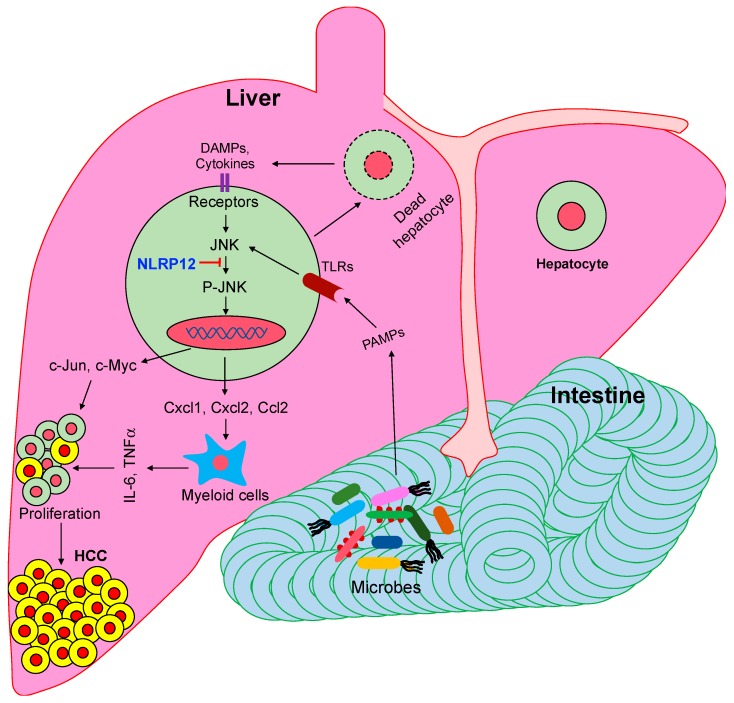
NLRP12 attenuates hepatocellular carcinoma (HCC) pathogenesis via suppression of pathogen-associated molecular patterns (PAMPs)-mediated activation of c-Jun N-terminal kinase (JNK). Gut-derived microbial pattern molecules transport to the liver where they are sensed by toll-like receptors (TLRs), leading to the activation of the JNK pathway. Cytokines, chemokines, and protooncogenes induced by the activated JNK promote the proliferation of neoplastic cells in the liver. NLRP12 inhibits JNK activation, and thereby suppresses HCC pathogenesis.

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
