# Peer review of "Crosstalk between NLRP12 and JNK during Hepatocellular Carcinoma"

_ijms, 2020, doi:10.3390/ijms21020496_

Round 1
Reviewer 1 Report
The first sentence "Hepatocellular carcinoma (HCC) is the fifth most common type of primary liver cancer" is incorrect. HCC is the most common form of primary liver cancer.
Author Response
We thank the reviewer for the time in reviewing our manuscript. We appreciate the comment pointing to an incorrect informaiton. We have rcorrected it in the revised manuscript. Please see the changes.

Reviewer 2 Report
The manuscript is well written and the illustrations are presented in a good quality. This will provide interesting information for the reader of the journal. Accordingly, I think the manuscript can be published in the present form.
Author Response
Many thanks for favorable comments and recommendation for publication.